# Characterization a Novel Butyric Acid-Producing Bacterium *Collinsella*
*aerofaciens* Subsp. *Shenzhenensis* Subsp. Nov.

**DOI:** 10.3390/microorganisms7030078

**Published:** 2019-03-13

**Authors:** Panpan Qin, Yuanqiang Zou, Ying Dai, Guangwen Luo, Xiaowei Zhang, Liang Xiao

**Affiliations:** 1BGI Education Center, University of Chinese Academy of Sciences, Shenzhen 518083, China; 2BGI-Shenzhen, Shenzhen 518083, China; zouyuanqiang@genomics.cn (Y.Z.); daiying@genomics.cn (Y.D.); luoguangwen@genomics.cn (G.L.); zhangxiaowei@genomics.cn (X.Z.); 3China National Genebank-Shenzhen, BGI-Shenzhen, Shenzhen 518120, China

**Keywords:** novel subspecies, *Collinsella*, phylogenetics, biochemistry, comparative genome

## Abstract

Butyrate-producing bacteria can biosynthesize butyrate and alleviate inflammatory diseases. However, few studies have reported that the genus *Collinsella* has the ability to produce butyric acid. Here, our study depicts a *Collinsella* strain, which is a rod-shaped obligate anaerobe that is able to produce butyric acid. This microorganism was isolated from a human gut, and the optimal growth conditions were found to be 37 °C on PYG medium with pH 6.5. The 16S rRNA gene sequence demonstrated that this microorganism shared 99.93% similarity with *C*. *aerofaciens* ATCC 25986^T^, which was higher than the threshold (98.65%) for differentiating two species. Digital DNA–DNA hybridization and average nucleotide identity values also supported that this microorganism belonged to the species *C. aerofaciens*. Distinct phenotypic characteristics between TF06-26 and the type strain of *C*. *aerofaciens*, such as the fermentation of D-lactose, D-fructose and D-maltose, positive growth under pH 5 and 0.2% (*w*/*v*) cholate, suggested this strain was a novel subspecies. Comparative genome analysis revealed that butyric acid kinase and phosphate butyryltransferase enzymes were coded exclusively by this strain, indicating a specific butyric acid-producing function of this *C*. *aerofaciens* subspecies within the genus *Collinsella*. Thus, *Collinsella*
*aerofaciens* subsp. *shenzhenensis* subsp. nov. was proposed, with set strain TF06-26^T^ (=CGMCC 1.5216^T^ = DSM 105138^T^) as the type strain.

## 1. Introduction 

The isolation and cultivation of microorganisms are rate-limiting steps in the study of microbiology. The study of intestinal bacteria isolation was boosted in the 1960s and 1970s by the emergence and use of anaerobic cultivation techniques [1]. The breakthroughs in modern sequencing technologies have enabled researchers to obtain more genomics information to identify new organisms. The number of isolated and cultivated gut microorganisms has increased considerably in recent years, especially butyric acid-producing bacteria. Butyrate, a product of intestinal microbial fermentation, plays an important role in colonic health and serves as an energy source for epithelial cells. Previous studies have suggested that butyric acid-producing bacteria could alleviate inflammatory bowel disease, type 2 diabetes and obesity [2,3,4].

In this study, a new strain of genus *Collinsella* is reported. Nine species in this genus, including *C. aerofaciens* [5], *C*. *intestinalis* [6], *C*.*tanakaei* [7], *C*.*stercoris* [6], *C*. *bouchesdurhonensis* [8], *C*. *phocaeensis* [9], *C*. *ihuae* [10], *C*. *provencensis* [11] and *C*. *vaginalis* [12], have been reported before. Several studies demonstrated that *Collinsella* could modify host bile acids and plasma cholesterol levels [13,14,15]. Acetic acid, formic acid and lactic acid are the main fermented products of the genus *Collinsella*. However, the function of producing butyric acid has not been reported yet. The objective of this study is to identify and characterize a new strain of a butyric acid-producing organism after isolating it from a human gut.

## 2. Materials and Methods

### 2.1. Organism Isolation and Cultivation 

The fecal sample used in this study was supplied by a 17-year-old healthy Chinese female volunteer. An approximately 500 mg sample was suspended in 1 mL of phosphate-buffered saline (PBS), and diluted serially in the same buffer. Next, 100 μL of the sample was spread on a peptone–yeast extract–glucose (PYG) agar plate [16]. After incubating at 37 °C for 3 days, colonies of bacteria were transferred and sub-cultured until a pure culture was obtained. The pure colony was preserved for further inoculation and characterization. All experimental operations were performed under anaerobic conditions (90% nitrogen, 5% hydrogen and 5% carbon dioxide, by volume).

### 2.2. Short Chain Fatty Acid-Producing Activity Assay

Butyric acid and other short chain fatty acids (acetic acid, formic acid, propionate, isobutyric acid, isovaleric acid, valeric acid, benzoic acid and lactic acid) were measured by gas chromatograph (GC-2014C, Shimadzu, Kagoshima, Japan) after fermenting the strains in PYG medium for 72 h. The capillary columns were packed with porapak HP-INNOWax (Cross-Linked PEG, 30 m × 0.25 mm × 0.25 µm) and maintained under 220 °C. N_2_ was used as the carrier gas in all analyses.

### 2.3. Genetic Identification

To determine the phylogenetic position of this strain, the 16S rRNA gene and the whole genome were sequenced. The 16S rRNA gene was amplified using a pair of universal primers, 27f (5′-AGAGTTTGATCATGGCTCAG-3′) and 1492r (5′-TAGGGTTACCTTGTTACGACTT-3′). RNAmmer was used to extend the length of the 16S rRNA gene [17]. The whole-genome sequence was obtained using the paired-end 100 bp strategy with the Illumina HiSeq 2000 at BGI-Shenzhen (Shenzhen, China). The raw data were filtered and assembled according to Zou’s method [16].

The 16S rRNA data were added to the EzBioCloud [18] database. Phylogenetic trees were constructed by the neighbor-joining [19] distance method using MEGA7 [20]. The stability of the relationships in the trees was assessed by bootstrapping 1000 replications. Digital DNA–DNA hybridization (DDH) and average nucleotide identity (ANI) were performed to determine the genetic relatedness between the isolated strain and members of the *Collinsella* species. Digital DDH was calculated by an online program, the Genome-to-Genome Distance Calculator (GGDC) (https://ggdc.dsmz.de/) [21]. ANI values were calculated by OrthoANI and ANIb [22,23].

### 2.4. Biochemical Characteristics Tests on PYG Medium

The effects of temperature, pH, NaCl and cholate on growth were tested in this study. The temperature ranged from 10 °C to 50 °C; pH was from 3 to 10; salinity ranged from 0 to 7% (*w*/*v*); cholate ranged from 0 to 0.5% (*w*/*v*). The pH was adjusted by adding NaOH or H_2_SO_4_.

Sugar metabolism was determined by commercial API 20A (bioMe’rieux) and API 50 CH tests (bioMe´rieux) according to the manufacturer’s instructions. Gram-stain reaction was tested according to Cappuccino and Sherman’s method in *Microbiology: A Laboratory Manual* (6th Edition). An antibiotic susceptibility test was also performed using the disc diffusion method in accordance with the manufacturer’s instructions.

### 2.5. Comparative Genome Analysis

A Bacterial Pan Genome Analysis (BPGA) pipeline [24] was used to compare genomes of TF06-26 and 13 strains of the genus *Collinsella*. The set of genes shared by all strains were defined as the core genome, while the global gene repertoire of all strains was defined as the pan genome. Genes partially shared in reference strains or unique to single reference strains have been defined as accessory genes and unique genes, respectively [25,26]. The phylogeny and function of core genes were also used in the BPGA pipeline by default parameters.

### 2.6. Data Availability

The GenBank/EMBL/DDBJ accession number for the 16S rRNA gene sequence of *Collinsella aerofaciens* TF06-26 is MF383464. The GenBank/EMBL/DDBJ accession number for the genome sequence of TF06-26 is NKXR00000000. Compliance with Ethical Standards: All procedures performed in studies involving human participants were in accordance with the ethical standards of BGI-IRB (The institutional review board on bioethics and biosafety of BGI) and with the 1964 Helsinki declaration and its later amendments or comparable ethical standards. Informed consent was obtained from all individual participants included in the study.

## 3. Results and Discussion

Through anaerobic isolation and cultivation, a rod-shaped (1.0 μm long and 1.0–4.0 μm wide), non-motile, non-spore-forming and Gram-positive obligate anaerobic microorganism was obtained. Gas chromatograph showed that 4.37 mmol/L butyric acid, 19.88 mmol/L acetic acid, 38.67 mmol/L lactic acid and 2.19 mmol/L benzoic acid were produced after 72 h of fermentation in PYG medium. TF06-26 could grow at 25 °C to 45 °C (optimum 37 °C) with pH 5.0–8.0 (optimum pH 7.0) as well as tolerate 0.3% (*w*/*v*) cholate and 2.0% (*w*/*v*) salt. The antibiotic susceptibility test demonstrated that this strain was resistant to kanamycin, amikacin and framycetin, whilst being sensitive towards ampicillin, carbenicillin and cefazolin. The general features of TF06-26 are showed in Table 1; more details of the growth conditions and antibiotic susceptibility test are shown in Appendix A.

To determine the phylogenetic position of TF06-26, the 1494 bp 16S rRNA gene and 2,260,546 bp genome were obtained. The 16S rRNA gene sequence was compared against the EzBioCloud database, and the sequence alignments showed that TF06-26 was phylogenetically close to the genus *Collinsella*, especially the species *C*. *aerofaciens*. TF06-26 exhibited 99.93%, 99.65% and 99.51% identity similarity with *C*. *aerofaciens* ATCC 25986^T^, *C*. *aerofaciens* indica and *C*. *aerofaciens* 2789STDY5834902, respectively, and all values were higher than the threshold (98.65%) for differentiating two species [27]. A dendrogram depicting the phylogenetic relationships of TF06-26 is shown in Figure 1. TF06-26 formed a distinct subline associated with *C*. sp. 4_8_47FAA (100.00% similarity), the uncultured bacterium clone TS27_a04b05 (96.25% similarity), *C*. *aerofaciens* ATCC 25986^T^, *C*. *aerofaciens* indica and *C*. *aerofaciens* 2789STDY5834902. These results suggest that it is likely that TF06-26 can be ascribed to the species *C*. *aerofaciens*.

Digital DDH relatedness and ANI were performed between TF06-26 and nine type strains of the genus *Collinsella* to verify the phylogenetic position of TF06-26. More details of these strains are shown in Appendix A. Consistent with the results of 16s rRNA gene similarity, both digital DDH relatedness and ANI values showed that TF06-26 is most closely related to the type strains of the species *C*. *aerofaciens* (Table 2). However, 65.7% DDH relatedness and 93.08% ANI values were slightly below the common threshold for the same species (70% DDH and 95% ANI). Considering the extremely high 16S rRNA similarity, four *C*. *aerofaciens* strains were added for comparison. These four strains were simultaneously assigned to *C*. *aerofaciens* by the NCBI genome database and the GTDB [28], which confirmed their phylogenetic positions. TF06-26 exhibited 71.19–75.9% digital DDH relatedness with these four *C*. *aerofaciens* strains, which were higher than the 70% threshold (Table 3). However, 93.25–95.05% OrthoANI values and 93.07–94.93% ANIb values within *C*. *aerofaciens* strains suggested that a 95% common threshold is not suitable for identifying this species, and 94.22% (TF06-26 with *C*. *aerofaciens* 2789STDY5834902), 94.13% (TF06-26 with *aerofaciens* indica) and 93.43% (TF06-26 with *C*. *aerofaciens* 2789STDY5608842) OrthoANI values were in the reasonable range among *C*. *aerofaciens* strains (Table 4 and Appendix A). Based on the above information, TF06-26 was assigned to the species *C*. *aerofaciens*.

The ad hoc committee for the reconciliation of approaches to bacterial systematics advised that bacterial isolates within a given species be considered as distinct subspecies if they differ phenotypically [29]. Carbon source metabolism and growth condition tests were undertaken to compare the phenotype characteristics between TF06-26 and the type strains of the species *C*. *aerofaciens*. Fermenting D-lactose, sucrose, salicin, D-galactose, D-fructose, D-mannose, arbutin, esculine, cellobiose, D-maltose and gluconate distinguished TF06-26 and *C*. *aerofaciens* ATCC 25986^T^. Additionally, TF06-26 exhibited a stronger ability to adapt to acidic and high-cholate environments with positive growth at pH 5.0 and 2% (*w*/*v*) cholate, in contrast to *C*. *aerofaciens* ATCC 25986^T^ (Table 5). Furthermore, TF06-26 was able to produce butyric acid and benzoic acid, and these distinct characteristics indicate that TF06-26 is a new subspecies of species *C*. *aerofaciens*.

As well as comparing strain types, this study also attempted to find different characteristics between TF06-26 and other strains of the species *C. aerofaciens*. Since limited phenotypic information of other *C. aerofaciens* strains has been reported, a comparative genome analysis was performed to further investigate genomic differences. Four non-type strains and eight additional type strains of the genus *Collinsella* were included. The pan-genome of these 14 genomes consisted of 535 core genes, 2622 accessory genes and 4160 unique genes (Appendix A, Appendix A). When compared with additional type strains, the *C. aerofaciens* strains and TF06-26 had more accessory genes and fewer unique genes. The dendrogram of both core genes and pan genes show that *C*. *aerofaciens indica* has the closest phylogenetic relationship with TF06-26 (Figure 2). The KEGG functional annotation of all 14 genomes indicates that there is a higher level of unique genes in carbohydrate metabolism function, and the same trend can be observed for core genes in translation function (Appendix A). Identifying COG functional categories showed that core genomes were enriched in the (J) category (translation, ribosomal structure and biogenesis), while accessory genes and unique genes could mostly be ascribed to the (R) category (general function prediction only), the (G) category (carbohydrate transport and metabolism) and the (K) category (transcription) (Appendix A). In our new strain TF06-26, 130 genes were specific. Functional annotation of these unique genes showed that butyric acid kinase (EC:2.7.2.7), phosphate butyryltransferase (EC:2.3.1.19), mannose-specific IIB component (EC:2.7.1.191) and mannose-specific IIC component were only coded by TF06-26 (Table 6). The two former enzymes are related to butyric acid biosynthesis and the two latter enzymes are part of the mannose transportation process. This suggests that only TF06-26 has the ability to produce butyric acid and fermentation mannose within the genus *Collinsella*.

## 4. Conclusions

In this study, a new butyric acid-producing *Collinsella* bacterium was tested after being isolated from a human gut. The morphology of this bacterium was Gram-positive, obligate anaerobic, non-motile, non-spore-forming and rod-shaped. The growing conditions of TF06-26 on PYG medium included a temperature from 25 °C to 45 °C (optimum 37 °C) with pH 5.0–8.0 (optimum pH 7.0). The 16S rRNA gene similarity, DDH relatedness values and ANI values all indicated that TF06-26 should be assigned to *C*. *aerofaciens*. The differing phenotypical characteristics, such as fermented D-lactose, D-fructose and D-maltose, biosynthesis butyric acid and its adaptability to acidic environments, proved that this microorganism is distant from the type strain of the species *C*. *aerofaciens*. Additionally, comparative genome analysis demonstrated genotypes unique from TF06-26 in butyric acid biosynthesis and mannose transportation within the species *C*. *aerofaciens*, even within the genus *Collinsella*. Thus, it is proposed that TF06-26 was classified as a novel subspecies of *C*. *aerofaciens* and named *Collinsella aerofaciens* subsp. *shenzhenensis* subsp. nov., in which TF06-26^T^ (=CGMCC 1.5216^T^ = DSM 105138^T^) was set as the type strain.

## Figures and Tables

**Figure 1 microorganisms-07-00078-f001:**
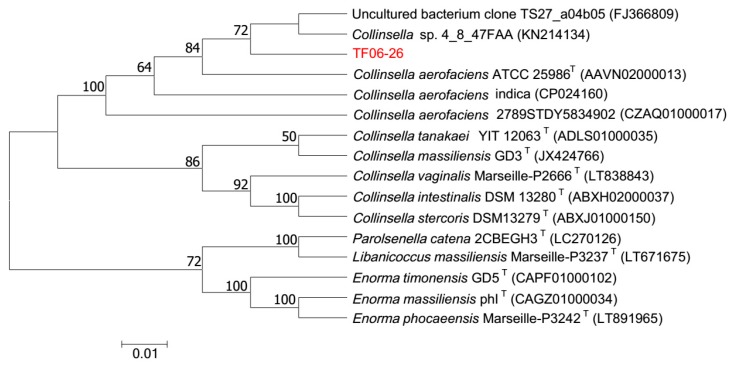
Neighbor-joining phylogenetic tree based on 16S rRNA gene sequences of C. aerofaciens TF06-26 and related type species. Bootstrap values based on 1000 replications are shown at branch nodes, and only noted if the percentage is greater than 50%. TF06-26 showed a close relationship with the genus *Collinsella*.

**Figure 2 microorganisms-07-00078-f002:**
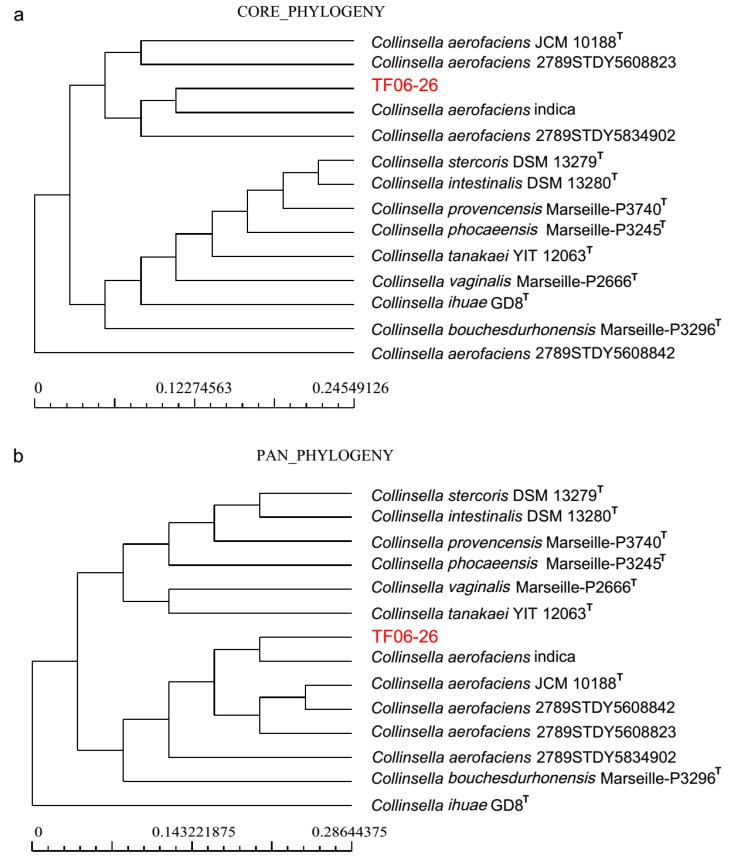
Neighbor-joining phylogenetic tree based on core genes and pan genes by Bacterial Pan Genome Analysis (BPGA) pipeline. (**a**) 535 core gene-based phylogenetic tree; (**b**) 7317 pan gene-based phylogenetic tree. TF06-26 showed a close relationship with the species *C. aerofaciens* and *C*. *aerofaciens indica* had the closest phylogenetic relationship with TF06-26.

**Table 1 microorganisms-07-00078-t001:** Parameters of strain TF06-26 concerning morphology, growth conditions, fermented production and genome information.

Characteristic	Parameter
Habitat	Human gut
Oxygen requirement	Anaerobic
Medium	Peptone–yeast extract–glucose (PYG) agar
Gram stain	Positive
Cell shape	Rod
Motility	Non-motile
Sporulation	Not reported
Biotic relationship	Free-living
Temperature range; optimum (°C)	25–45; 37
pH range; optimum	5–8; 7
Salinity (*w*/*v*)	0–2%
Cholate toleration (*w*/*v*)	0–0.3%
Butyrate (mmol/L)	4.37
Acetic acid (mmol/L)	19.88
Lactic acid (mmol/L)	38.67
Benzoic acid (mmol/L)	2.19
GenBank accession of 16s rRNA	MF383464
GenBank accession of genome	NKXR00000000
DNA G + C (%)	59.81

**Table 2 microorganisms-07-00078-t002:** The digital DNA–DNA hybridization (DDH), Ortho-average nucleotide identity (ANI) and ANIb values between TF06-26 and the representative strains of other nine species of the genus *Collinsella.*

Strains	DDH (%)	OrthoANI (%)	ANIb (%)
*C*. *aerofaciens* ATCC 25986 ^T^	65.7	93.08	92.92
*C*. *bouchesdurhonensis* Marseille-P3296 ^T^	16.1	76.18	75.12
*C*. *intestinalis* DSM 13280 ^T^	15.3	75.24	74.93
*C*. *phocaeensis* Marseille-P3245 ^T^	16.5	75.64	74.64
*C*. *stercoris* DSM 13279 ^T^	15.8	76.37	75.38
*C*. *tanakaei* YIT 12063 ^T^	15.2	75.34	73.83
*C*. *ihuae* GD8 ^T^	15.7	75.48	74.62
*C*. *provencensis* Marseille-P3740 ^T^	14.9	74.40	73.90
*C*. *vaginalis* Marseille-P2666 ^T^	14.4	74.82	73.60

^T^: Type strain.

**Table 3 microorganisms-07-00078-t003:** The digital DDH values between TF06-26 and five strains of the species *C*. *aerofaciens.*

Strains	1	2	3	4	5	6
1	−	75.9	74.2	71.3	71.1	65.7
2	75.9	−	74.6	74.4	72.3	70
3	74.2	74.6	−	72.2	73.5	69.4
4	71.3	74.4	72.2	−	79	78.4
5	71.1	72.3	73.5	79	−	73.6
6	65.7	70	69.4	78.4	73.6	−

1, TF06-26; 2, *C*. *aerofaciens* 2789STDY5834902; 3, *C*. *aerofaciens* indica; 4, *C*. *aerofaciens* 2789STDY5608842; 5, *C*. *aerofaciens* 2789STDY5608823; 6, *C*. *aerofaciens* ATCC 25986 ^T^; ^T^: Type strain.

**Table 4 microorganisms-07-00078-t004:** The OrthoANI values between TF06-26 and five strains of the species *C*. *aerofaciens.*

	1	2	3	4	5	6
1	−	94.22	94.13	93.43	93.14	93.08
2	94.22	−	93.71	93.38	93.29	93.25
3	94.13	93.71	−	93.76	93.73	93.58
4	93.43	93.38	93.76	-	95.05	94.83
5	93.14	93.29	93.73	95.05	−	94.61
6	93.08	93.25	93.71	94.83	94.61	−

1, TF06-26; 2, *C*. *aerofaciens* 2789STDY5834902; 3, *C*. *aerofaciens* indica; 4, *C*. *aerofaciens* 2789STDY5608842; 5, *C*. *aerofaciens* 2789STDY5608823; 6, *C*. *aerofaciens* ATCC 25986^T^; ^T^: Type strain.

**Table 5 microorganisms-07-00078-t005:** Characteristics that differentiate TF06-26 and *C. aerofaciens* ATCC 25986 ^T^.

Test	Characteristic	TF06-26	*C*. *aerofaciens* ATCC 25986
API 20A	D-Lactose	+	w
	Sucrose	−	+
	Salicin	w	−
API 50 CH	D-Galactose	+	w
	D-fructose	+	w
	D-mannose	+	w
	Arbutin	w	−
	Esculine	+	−
	Salicin	w	−
	Cellobiose	+	−
	D-Maltose	+	w
	D-Lactose	+	w
	D-Sucrose	−	w
	Gluconate	−	w
pH	5	+	−
	5.5	+	−
	7.5	+	++
	8.5	−	w
Bile % (*w/v*)	0.2	+	w
Butyrate		Yes	No
Benzoic acid		Yes	No

+, positive; w, weakly positive; –, negative; ^T^: Type strain.

**Table 6 microorganisms-07-00078-t006:** The KO annotation of the unique genes of TF06-26.

KO	Definition	Score
K00634	ptb; phosphate butyryltransferase (EC:2.3.1.19)	194
K02795	PTS-Man-EIIC; PTS system, mannose-specific IIC component	293
K02794	PTS-Man-EIIB; PTS system, mannose-specific IIB component (EC:2.7.1.191)	198
K12111	ebgA; evolved beta-galactosidase subunit alpha (EC:3.2.1.23)	1024
K03475	PTS-Ula-EIIC; PTS system, ascorbate-specific IIC component	544
K22044	ybiO; moderate conductance mechanosensitive channel	325
K19545	lnuA_C_D_E; lincosamide nucleotidyltransferase A/C/D/E	303
K06941	rlmN; 23S rRNA (adenine2503-C2)-methyltransferase (EC:2.1.1.192)	298
K00847	E2.7.1.4; fructokinase (EC:2.7.1.4)	291
K03655	recG; ATP-dependent DNA helicase RecG (EC:3.6.4.12)	249
K05946	tagA; N-acetylglucosaminyldiphosphoundecaprenol N-acetyl-beta-D-mannosaminyltransferase (EC:2.4.1.187)	215
K00817	hisC; histidinol-phosphate aminotransferase (EC:2.6.1.9)	215

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
