# Peer review of "Characterization a Novel Butyric Acid-Producing Bacterium Collinsella aerofaciens Subsp. Shenzhenensis Subsp. Nov."

_microorganisms, 2019, doi:10.3390/microorganisms7030078_

Reviewer 1 Report

Dear authors,

please receive here below my comments on your manuscript "Characterization a butyric acid-producing novel bacterium Collinsella aerofaciens  subsp. shenzhenensis  subsp. nov."

In general, the manuscript is well structured and the conclusions are well justified by the results.

Please, highlight more why a human gut butyric acid-producing bacterium is important (this is described in the Introduction, but not in the abstract).

English is well written but is weird at some locations.

Please find here below my detailed comments:

Title: please write "Characterization of a novel butyric acid-producing bacterium Collinsella aerofaciens  subsp. shenzhenensis  subsp. nov."

Authors: please modify the symbol "cross" symbol. It can be confused as a posthumous symbol. Please use a different symbol, e.g. "§"

Abstract: Please highlight why a human gut butyric acid-producing bacterium is important.

Introduction: Page2Line60-61. Please rewrite the full sentence as: "The objective of this study is the identification and characterization of an isolated butyric acid-producing Collinsella organism.

Page3Line89. Please describe the media used for the biochemical characteristics tests. If you have used the same media as in "short chain fatty acid production" please precise that.

Page3Line105. "... was isolated from the fecal of a 17 year old healthy Chinese female." this was already said in the materials and methods, please get rid of this. 

Table 1. Please correct the editing, Table 1 caption is separated from the table, and manuscript text is too close to Table 1.

Page5Line141: please write "it was compared..." instead of "...we then compared..."

Page5Line146: Please write "it was used..." instead of "we used two..."

Page6Line170: please write "distant" instead of "distance"

Page7Line189: Please revise English "... and may only this strain can." 

Page10Line206: please write "It was isolated from human faeces" instead of "it was isolated from the human faeces".

Author Response

Thanks for the revision of this manuscript, especially the advice of symbol.

 Point 1: Highlight more why a human gut butyric acid-producing bacterium is important (this is described in the Introduction, but not in the abstract). 

 Response 1: Revision sequence “Collinsella is a genus of Actinobacteria, known for their ability to ferment a wide range of carbohydrates, forming products such as acetic acid, formic acid and lactic acid, but forming butyric acid has not been reported.” to “Butyrate-producing bacterium can biosynthesize butyrate and alleviate inflammatory disease. However, barely studies reported that genus Collinsella has the ability to produce­­­ butyric acid.” in abstract.

 Point 2: English is well written but is weird at some locations.

 Response 2: Thank you for your grammatical suggestions, we have sought help from a native English speaker.

Reviewer 2 Report

Here in the manuscript, Qin et al. describe a novel butyric acid-producing Collinsella aerofaciens.

The paper is important, as gut butyric acid producing bacteria have in the past been shown to promote health. However, I have reserves about the paper in the current state. First, the grammar throughout the manuscript should be significantly improved. Second, the authors propose a new subspecies shenzhenensis (strain TF06-26) within Collinsella aerofaciens. They base this claim on DDH 65.7% and orthoANI 93.25 (strain TF06-26 compared to C. aerofaciens type strain). DDH values to other strains of the species was from 71.1 to 75.9 %   and ANI values ranged from 93.14 to 94.22. As far as I am aware, the current guidelines for species delineation propose DDH above 70% and ANI above 95. The guidelines for subspecies delineation propose the following: 1) OGRIs (DDH and ANI) between subspecies and other species should be lower than the species-level cutoff value, 2) OGRIs between subspecies should be higher than the species-level cutoff, 3) strains belonging to different subspecies should be genomically coherent and form distinguishable clades by OGRIs and phylogenomic treeing. This basically means that the numbers of DDH and ANI values when comparing the proposed subspecies to other C. aerofaciens species should be above 70% and 95, respectively. The next problem is that the authors have only one strain and therefore we cannot really check the third point. Also the authors have only compared the phenotypic characteristics of the new strain TF06 – 26 and the type strain of the species C. aerofaciens. This is problematic as the DDH and ANI numbers suggest that the type strain may not be a typical representative of the species. Therefore, it is my opinion that they should also have tested the other strains of the species. The data in my opinion also suggest that the assignment of some of the strains to the C. aerofaciens species should be reassessed.

Author Response

Thanks for the revision of this manuscript, your question about the

Thanks for the revision of this manuscript, your question about the phylogenetic position of this new bacteria is also a difficult point in our analysis.

 Point 1: First, the grammar throughout the manuscript should be significantly improved.  

 Response 1: We sought help from a native English speaker and modified plenty sentence

 Point 2: Is it correct of the phylogenetic position of TF06-26?

 Response 2: Thanks for given the guidelines for subspecies delineation. We think the key to this problem is the threshold for same species. In this manuscript,

1)      TF06-26 shared 99.93 % 16s similarity with C. aerofaciens ATCC 25986 T, higher than the threshold (98.65%) for differentiating two species and the similarity with other two strains of species also higher than 99%

2)      TF06-26 exhibited 71.19%-75.9% digital DDH relatedness with these 4 C. aerofaciens strains, which were higher than the 70% threshold.

3)      As you proposed, the ANI values were not reached the 95% threshold. But 93.25%-95.05% OrthoANI values and 93.07%-94.93% ANIb values within all C. aerofaciens strains suggested 95% common threshold is not suitable for this species, and 94.22% (TF06-26 with C. aerofaciens 2789STDY5834902), 94.13% (TF06-26 with aerofaciens indica) and 93.43% (TF06-26 with C. aerofaciens 2789STDY5608842) OrthoANI values were were in the reasonable range among C. aerofaciens strains.

In addition, these 4 C. aerofaciens strains used for analysis were simultaneously assigned to C. aerofaciens by NCBI genome database and GTDB taxonomy database which ensure its correction of phylogenetic position. So we think this new strain belongs to specie C.aerofaciens. And the ad hoc committee on reconciliation of approaches to bacterial systematics was proposed that bacterial isolates within a given species could be considered as distinct subspecies if they are differed phenotypically, so based on those distant characteristics F06-26 was proposed as a new subspecies of species C. aerofaciens

 Point 3: only compared the phenotypic characteristics of the new strain TF06-26 and the type strain of the species C. aerofaciens.

 Yes, you are right, because of lacking reported phenotypic information of other C. aerofaciens strains, we only compared the phenotypic characteristics with the type strain. But we used genomic comparative analysis to deeply investigate genomic differences. Though the unique genes of TF06-26, it was inferred genotypic unique of TF06-26 in butyric acid biosynthesis and mannose transportation within species C. aerofaciens, even within genus Collinsella.

 Round  2

Reviewer 2 Report

I find the revised manuscript signifficantly improved with authors reponding to all the major issues.

The english should still be improved (grammatical and spelling errors) throughout the manuscript.

Author Response

Thank you very much, your question has also helped us sort out the ideas. Spelling and grammar were re-revised.